# The Impact on Carbon Emissions of China with the Trade Situation versus the U.S.

**Jieming Chou** [1], **Fan Yang** [1,2,*], **Zhongxiu Wang** [3] **and Wenjie Dong** [4]

1   State Key Laboratory of Earth Surface Processes and Resource Ecology, Faculty of Geographical Science, Beijing Normal University, Beijing 100875, China; choujm@bnu.edu.cn
2   CMA Key Laboratory for Cloud Physics of China Meteorological Administration, Chinese Academy of Meteorological Sciences, Beijing 100081, China
3   The Alliance of International Science Organizations, Beijing 100085, China; wang_zhongxiu@126.com
4   Southern Marine Science and Engineering Guangdong Laboratory (Zhuhai), Zhuhai 519080, China; dongwj3@mail.sysu.edu.cn
*   Correspondence: yangfan_bnu@mail.bnu.edu.cn; Tel.: +86-189-1052-3251

**Abstract:** The China–US trade conflict will inevitably have a negative impact on China's trade imports and exports, industrial development, and economic growth, and will affect the achievement of climate change goals. In the short term, the impact of the trade conflict on China's import and export trade will cause the carbon emissions contained in traded commodities to change accordingly. To assess the impact of the trade conflict on China's climate policy, this paper combines a model from the Global Trade Analysis Project (GTAP) and the input–output analysis method and calculates the carbon emissions in international trade before and after the conflict. The conclusions are as follows: (1) The trade war has led to a sharp decline in China–US trade, but for China as a whole, imports and exports have not changed much; (2) China's export emissions have changed little, its import emissions have dropped slightly, and its net emissions have increased; and (3) China's exports are still concentrated in energy-intensive industries. Changes in trade will bring challenges to China's balancing of climate and trade exigencies. China–US cooperation based on energy and technology will help China cope with climate change after the trade conflict.

**Keywords:** trade conflict; carbon emissions; import and export trade; cooperative emission reduction

## 1. Introduction

Currently, the world economy suffered unexpected shocks [1], affected by the epidemic COVID-19 [2]. The United States and China are the two largest economies: China relied on its institutional advantages to control the number of domestic cases [3] and the economy recovered rapidly in the US due to the popularization of vaccines. American citizens are eager for excess savings during the retaliatory consumption epidemic, and many industries are experiencing inflation [4]. Among their major suppliers, in addition to China, countries in south and southeast Asia are hardest hit by the epidemic, and it is even difficult for India to control its own situation [5,6]. The trade tensions between China and the United States tend to ease, and have been an important factor affecting international trade in recent years.

In July 2018, the United States began to impose 25% tariffs on an array of Chinese exports worth US $34 billion [7], and China and the United States began a trade war that has had an enormous impact on the economic development of the two countries as well as the world economy and global trade [8,9]. The strategic conflict between China and the US emerged at the end of 2017, when China was portrayed as a competitor in a Trump administration National Security Report [10]. The trade conflict between China and the United States reflects the strategic competition between the two countries in the new industrial revolution. In turn, future trade agreements may be conditioned on climate agreements in international negotiations. Biden's presidential campaign plan called for

binding agreements on enhanced climate ambition, including shipping and aviation, and Biden may support the adoption of a carbon border adjustment [11,12].

Economic growth and rapid industrialization are considered to be the main reasons for the sharp increase in emissions [13]. Since 2006, China has been the world's largest carbon emitter [14]. At the same time, China is also the largest net exporter of carbon dioxide emissions in goods and services [15,16]. The increase in emissions embodied in China's trade has caused problems for international trade and climate policy: China and other emerging markets have a comparative advantage in manufacturing and are an essential part of international trade; however, at the same time, because China's carbon-intensive manufacturing yields much more carbon emissions than the manufacture of the same products in developed regions, trade has increased global carbon emissions [17–20].

With the rapid development of international trade, the production chain of goods and services is no longer limited to one or two countries, and more production and consumption take place in different countries. Current accounting schemes for carbon emissions are mainly based on emissions from production, with less consideration of the consumption side [21,22]. There are two principal methods for consumption-based carbon emissions accounting: life cycle assessment (LCA) and input–output analysis (IOA) [23–25]. The LCA method is typically used for relatively simple and traceable inspections of production chains such as households and enterprises. On the other hand, input–output analysis is widely used at the national and sector levels [26–28]. This method can be further divided into three model frameworks: single region input–output (SRIO), bilateral trade input–output (BTIO), and multi-regional input–output (MRIO).

The SRIO model is mostly used to study the implied energy and emissions in a country's trade, taking the country as a whole and assuming the same production technology; the BTIO model takes into account technological differences between different countries and uses separate energy consumption and emissions factors; neither of these two methods can accurately reflect the relationship between industry and trade among various sectors in each country [29]. The MRIO model distinguishes between the technical and economic structures of different countries as well as the flow of imported and exported products [30]. With the improvement of input–output tables among countries, this method is increasingly employed in research on large-scale hidden emissions in global trade. In its trade war simulation, this paper mainly focuses on changes in China's trade and the resulting changes in emissions. The single region input–output model can meet the paper's research needs with fewer data requirements than the other models, so the SRIO model is adopted.

Here, we combine existing methods to simulate the impact of the trade conflict on China's commodity trade value [31,32] and to discuss the impact on China's energy industry and the path of carbon reduction. In order to track global import and export changes caused by trade conflicts, we use the model of Global Trade Analysis Project (GTAP) [33] to simulate the trade situation of 29 sectors in 14 regions. We calculate the emissions embodied in China's trade by a single input–output (SRIO) model of emissions and trade as of the year 2018. Our calculations only include carbon emissions from China's imports and exports, and emissions from other regions are not included.

## 2. Materials and Methods

### 2.1. Materials and Data

The GTAP model data are from the GTAP v10 data package [34], which contains the input–output tables and trade volumes of countries across the world. This paper uses a recursive method to project the 2014 data in the model to 2018 [35], and the currency is US dollars. The energy statistics for China's carbon emissions accounting come from the Energy Statistics Yearbook [36–40], and the emissions factors are derived from the revised emissions factors in Liu's study [41]. Due to the slow updating of China's statistical data, energy statistics for 2018 have not been released, so energy data of 2017 are used to generate carbon emissions data. At the same time, due to the difficulty of obtaining foreign data, this paper combines the emissions data contained in the GTAP's own database and

assumes that foreign countries in each region have similar technical levels and are unified into the same emissions coefficient matrix. Abbreviations for regions and departments can be found in Tables A1 and A2.

*2.2. Methods*

2.2.1. The GTAP Model

The model from the Global Trade Analysis Project (GTAP) is a multi-country multi-sector application general equilibrium model designed based on neoclassical economic theory (Hertel, 1997; GTAP, 2019; Walmsley et al., 2012). The GTAP, led by Thomas W. Hertel, a professor at Purdue University in the United States, was developed and has been widely used in the analysis of trade policies. In the GTAP model framework, they first establish a sub-model that can describe in detail the behavior of each country's production, consumption, government expenditure, etc., and then link the sub-models into a multi-country multi-sector general equilibrium model through international commodity trade. When we carry out policy simulations in this model framework, it is possible to simultaneously discuss the impact of the policy on factors such as production, imports and exports, commodity prices, factor supply and demand, factor compensation, gross domestic product, and social welfare levels in various countries.

The GTAP model assumes that the market is perfectly competitive, the returns to scale of production are constant, producers minimize production costs, consumers maximize utility, and all product and input factor markets clear. At the same time, each country has only one account, and all taxes, financial assets, and capital and labor income are accumulated in this account. The income in the account is divided into three parts: private consumption, deposits, and government consumption. The private expenditure equation uses the fixed difference elastic utility equation. The government's utility equation takes the form of a Cobb-Douglas equation.

GTAP establishes connections between countries (regions) through trade. Domestic products and imported products from different regions are incomplete substitutes; that is, they follow the Armington hypothesis and are characterized by a set constant elasticity of substitution. When the construction of a country's economic model is completed, the commodities and capital flows of international trade (the "global banking" sector) are added to it to form a multi-country economic model. At this time, there is a substitution relationship between imported products and domestic products, and the Armington hypothesis is adopted for product compounding; that is, imported products and domestic products are regarded as different products, and they have an incomplete substitution relationship between each other.

In the GTAP model, there are two international departments (national banks and international transportation departments). The savings of each country are aggregated into international banks and distributed among the countries according to the return on capital. The price expression of import and export commodities in the GTAP model is as follows:

$$P^{FOB} = P^{EX}\left(1 + T^{EX}\right) \tag{1}$$

$$P^{CIF} = P^{FOB}(1 + F) \tag{2}$$

$$P^{IM} = P^{CIF}\left(1 + T^{IM}\right) \tag{3}$$

where $P^{FOB}$ represents the export port price, $P^{EX}$ represents the domestic price of exported goods, $P^{CIF}$ represents the import port price, $P^{CIF}$ represents the domestic price of exported goods, $P^{IM}$ represents the domestic price of imported goods, $T^{EX}$ and $T^{IM}$ represent export and import tariffs (or subsidies), and $F$ is the freight cost.

### 2.2.2. Production-Based Carbon Emissions Calculation

We calculate the production-based emission according to the IPCC sectoral approach [41]. Emissions are calculated based on the sectoral consumption of different fuels, as shown in equation below.

$$CE_{ij} = AD_{ij} \times NCV_i \times CC_i \times O_{ij} \tag{4}$$

where $CE_{ij}$ refers to the carbon dioxide emissions generated by the combustion of fossil fuel type $i$ in sector $j$; $AD_{ij}$ represents the fossil fuel consumption of the corresponding type and sector; $NCV_i$ refers to the net calorific value, i.e., the calorific value generated by each fossil fuel combustion unit; $CC_i$ refers to the $CO_2$ emissions per unit of net calorific value generated by fossil fuel $i$; and $O_{ij}$ refers to the oxygenation efficiency. The fossil fuel emissions factors ($NCV_i \times CC_i$) we adopted are from a study by Liu [41], in which 602 groups of coal samples from all coal mines in China were sampled and weighted to obtain the national average emissions factor. Reference values for emission factors can be found in Table A3.

### 2.2.3. Input–Output Method to Calculate Trade Emissions

One method of consumption-based carbon emissions accounting is to compile an inventory based on the final consumption location of goods and services, and another including the total amount of the emissions contained in the imports used in production, and subtract the two quantities. The emissions included in exports reflect the interregional exchange of energy supply, commodities, and materials. Environmentally extended input–output analysis (EIO) can be used to calculate the emissions from regional imports and exports.

Input–output analysis is a method used to study the production balance among various sectors of the national economy. If we start from the assumption of general equilibrium, the dependence of the product volume of each sector is expressed as a system of equations. Then, based on statistical data, a matrix or checkerboard-shaped balance table is made to show the overall picture of the balance between the supply of and demand for products in various sectors of the national economy; from this is derived the total amount of products in each sector. The proportion of the product volume required by other sectors (called the technical coefficient) is used to determine the relevant parameter values in the above equations.

According to Leontief's input–output analysis method [42], the following models can be established:

$$X = AX + Y \tag{5}$$

where $X$ is the N*1 order total output column vector, N is the number of economic sectors, $Y$ is the N*1 order final product column vector, and matrix $A$ is the direct consumption coefficient.

After conversion, it can be transformed into:

$$X = (I - A)^{-1}Y = BY \tag{6}$$

Here, $B$ is the Leontief inverse matrix, that is, the complete demand coefficient matrix, and $I$ is the identity matrix.

Next, we can obtain the demand coefficient matrix $C$ of carbon emissions in each industry,

$$C = X^C(1 - A)^{-1} \tag{7}$$

where $X^C$ represents the carbon emissions on the production side of each sector.

Finally, we can obtain the carbon emissions in import and export trade,

$$C^{im} = CY^{im} \tag{8}$$

$$C^{out} = CY^{out} \tag{9}$$

where $C^{im}$ and $C^{out}$ represent the carbon emissions contained in imports and in exports, respectively, and $Y^{im}$ and $Y^{out}$ represent the import and export trade volumes, respectively.

## 3. Results

### 3.1. Goods Traded before and after the Trade Conflict

The model used in this paper is the GTAP model developed by researchers at Purdue University in the United States. It is a multi-country, multisector computable general equilibrium model and is widely used in quantitative analyses of the impact of international trade policies.

The trade conflict model setting reflects a scenario in which the United States imposes tariffs on different trade commodities to eliminate the trade deficit, and China counters with tariffs of its own. We run our simulations based on the list of 25% tariffs imposed on several key sectors. Changes in macroeconomic variables such as commodity trade variables in the process are the result of China's response to the impact of the trade war. Given the uncertainties surrounding different national policies, no scenario analysis was performed on this basis for other countries' policies (such as the EU's countermeasures to the US's increase in tariffs, countries around the world speeding up RCEP negotiations, etc.).

Table 1 shows the impact of the trade war on China's exports in various sectors. It can be seen that China's exports to the United States have fallen sharply, but its exports to other countries have increased. The total exports of most sectors have increased, mechanically leading to an increase in emissions from China's trade.

**Table 1.** Changes in China's exports to different countries.

| Sectors | USA | Oceania | EastAsia | SEAsia | SouthAsia | Namerica | La-Amer | EU-28 | MENA | SSA | Other |
|---|---|---|---|---|---|---|---|---|---|---|---|
| Transport | −0.74 | 0.06 | 0.05 | 0.05 | 0.07 | 0.00 | 0.05 | 0.06 | 0.06 | 0.06 | 0.06 |
| ElectricalEq | −0.07 | −0.05 | −0.06 | −0.06 | −0.04 | −0.13 | −0.06 | −0.06 | −0.05 | −0.05 | −0.05 |
| ElectronicEq | −0.15 | −0.07 | −0.07 | −0.08 | −0.06 | −0.14 | −0.08 | −0.07 | −0.07 | −0.07 | −0.07 |
| FerrousMetal | −0.02 | −0.03 | −0.04 | −0.04 | −0.03 | −0.09 | −0.04 | −0.03 | −0.03 | −0.03 | −0.03 |
| Total | −0.33 | 0.00 | −0.04 | −0.04 | −0.01 | −0.05 | 0.00 | −0.02 | 0.00 | 0.00 | −0.02 |

Table 2 shows the impact of the trade war on China's imports in various sectors. It can be seen that overall imports have been slightly reduced, and the changes are not very different across the various sectors. Imports from the United States and North America have changed significantly, mainly due to the increase in import costs caused by tariffs. Under the influence of this trend, Chinese imports from other countries have also been slightly reduced, mechanically leading to a reduction in the emissions contained in China's imported products. If we take the two together, China's trade exports have increased while its imports have decreased, and China's consumption-based carbon emissions have decreased in turn.

**Table 2.** Changes in China's imports from different countries.

| Sectors | USA | Oceania | EastAsia | SEAsia | SouthAsia | Namerica | LatinAmer | EU-28 | MENA | SSA | Other |
|---|---|---|---|---|---|---|---|---|---|---|---|
| GrainsFesFis | −0.67 | 0.09 | 0.10 | 0.09 | 0.10 | 0.06 | 0.09 | 0.09 | 0.10 | 0.10 | 0.10 |
| ProcFood | −0.66 | 0.01 | 0.01 | 0.00 | 0.02 | −0.04 | 0.01 | 0.01 | 0.01 | 0.01 | 0.01 |
| Transport | −0.74 | 0.06 | 0.05 | 0.05 | 0.07 | 0.00 | 0.05 | 0.06 | 0.06 | 0.06 | 0.06 |
| ChemicalPro | −0.76 | 0.02 | 0.02 | 0.01 | 0.03 | −0.03 | 0.02 | 0.02 | 0.03 | 0.03 | 0.03 |
| Total | −0.33 | 0.00 | −0.04 | −0.04 | −0.01 | −0.05 | 0.00 | −0.02 | 0.00 | 0.00 | −0.02 |

### 3.2. Carbon Emissions Contained in China's Trade with the US

According to the value of international trade and the emissions coefficient matrix, we calculate the emissions changes in these main sectors for the two countries and the emissions changes of other sectors.

In Figure 1a, it can be seen that the carbon emissions contained in goods imported from the United States by China in several major sectors have been reduced. Due to the

difference in carbon emissions intensity, the changes in emissions contained in chemical products are obviously greater, and the decreasing imports from other sectors also have the effect of decreasing the emissions contained in those sectors. As seen in Figure 1b, the changes in carbon emissions from China's exports to the United States are different from the changes in emissions from imports. Except for those of the nonferrous metal sector, the carbon emissions of sectors with tariffs are all relatively low, while the emissions of other sectors have increased by a large margin. This is similar to the result of the trade analysis. The import shrinkage effect caused by the trade conflict has mechanically reduced China's import emissions from the United States. However, at the same time, export emissions are controlled by the trade market and have grown slightly in other sectors that do not levy tariffs, with only small changes overall. On the whole, China's net emissions to the United States have decreased.

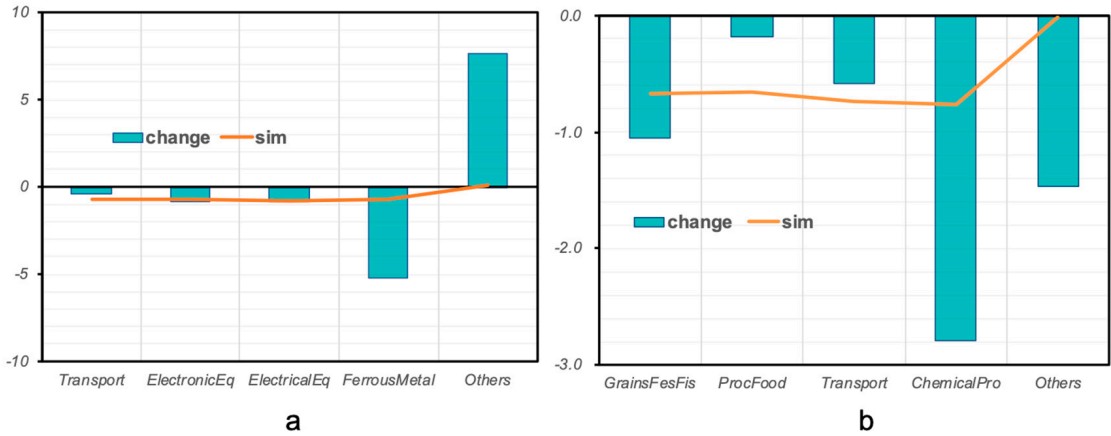

**Figure 1.** Changes in commodity carbon emissions from major sectors (MtCO$_2$) before and after the China–US trade conflict, where (**a**) represents emissions from China to the United States and (**b**) represents emissions from the United States to China.

### 3.3. Changes in China's Trade Emissions with the Rest of the World

Figures 2 and A1 shows the changes in China's export emissions to various countries in the world. Figure 2a shows the absolute change, and Figure 2b shows the percentage change. On the whole, China's exports to the world are mainly concentrated in the industrial and service industries at this stage, while the sectors with the largest export emissions are the electricity and water sectors, with emissions that are much higher than those of other sectors. Since the start of the trade conflict, except for in a few major sectors in which tariffs have been imposed, emissions have decreased, and those of other sectors have increased slightly.

Figures 3 and A2 shows the changes in emissions from China's imports from various countries in the world. Figure 3a shows the absolute change, and Figure 3b shows the percentage change. It can be seen that the distribution of emissions from China's imports is relatively even, with transportation services accounting for the largest share. Since the start of the trade conflict, the import emissions of all sectors have fallen, and China's import trade has been more affected.

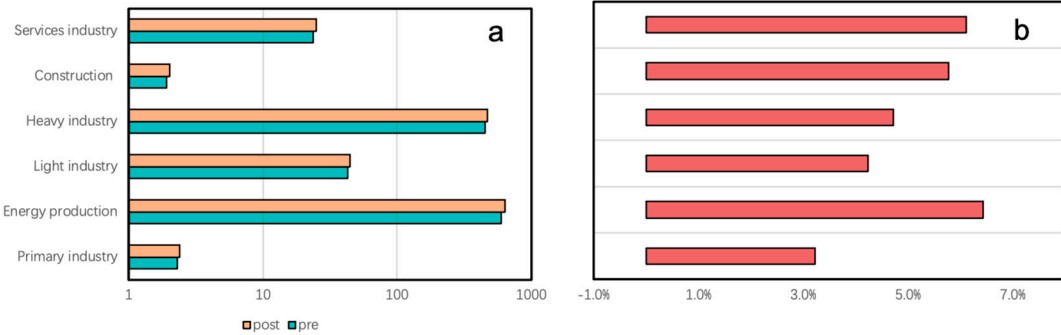

**Figure 2.** Changes in the world's carbon emissions from China's exports, where (**a**) represents the change in carbon emissions (MtCO$_2$) and (**b**) represents the percentage change.

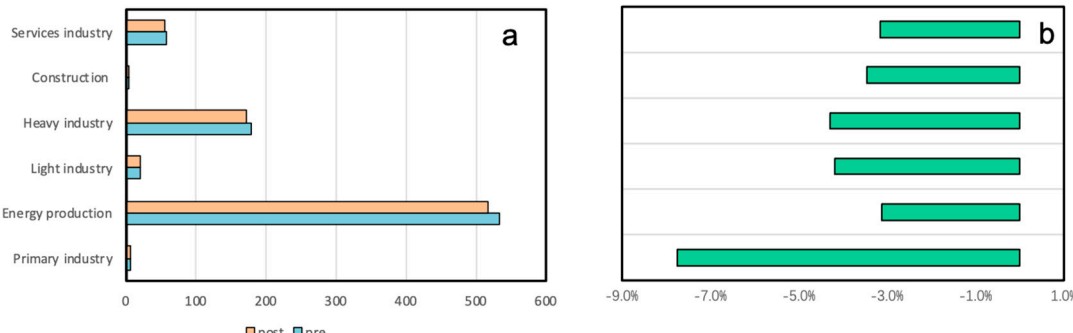

**Figure 3.** Changes in the world's carbon emissions from China's imports, where (**a**) represents the change in carbon emissions (MtCO$_2$) and (**b**) represents the percentage change.

## 4. Discussion

### 4.1. Spatial Distribution of Emissions Included in China's Trade

To further discuss the impact of the trade conflict on carbon emissions, this paper examines the changes in China's import and export emissions from different countries from a spatial perspective. As seen in Figure 4, whether through imports or exports, China's share of carbon emissions to the United States is smaller than the shares of other Asian countries. Due to the influence of spatial location, the countries that trade most with China are Asian countries. Whether because of transportation costs or the demand for a large number of daily necessities caused by population growth, these countries have closer trade ties with China. In contrast, China–US trade is more concentrated in certain sectors. Before the trade conflict, China's exports to the United States were electronic products, which accounted for 1/3 of all of China's exports and half of China's total exports of electronic products. Since the start of the trade conflict, the share has plummeted to approximately 1/8. On the other hand, the emissions coefficient of electronic products is so low that even before the trade conflict, the carbon emissions of electronic products accounted for only 1/50 of China's total emissions from exports to the United States.

Unlike China's exports to the United States, China's imports from the United States are the main component of the changes to China's imports. Compared with the emissions from imports from other countries and regions, which have shown only minor changes, China's emissions from imports from the United States have been reduced by nearly one-third, which has had an impact on China's overall import situation. Although China's import market is not highly dependent on the United States, the United States is an important source of imports for Chinese agricultural products and transportation equipment. China's response to the tariffs has also had a considerable impact on these two sectors, which have seen their imports reduced by nearly 70%.

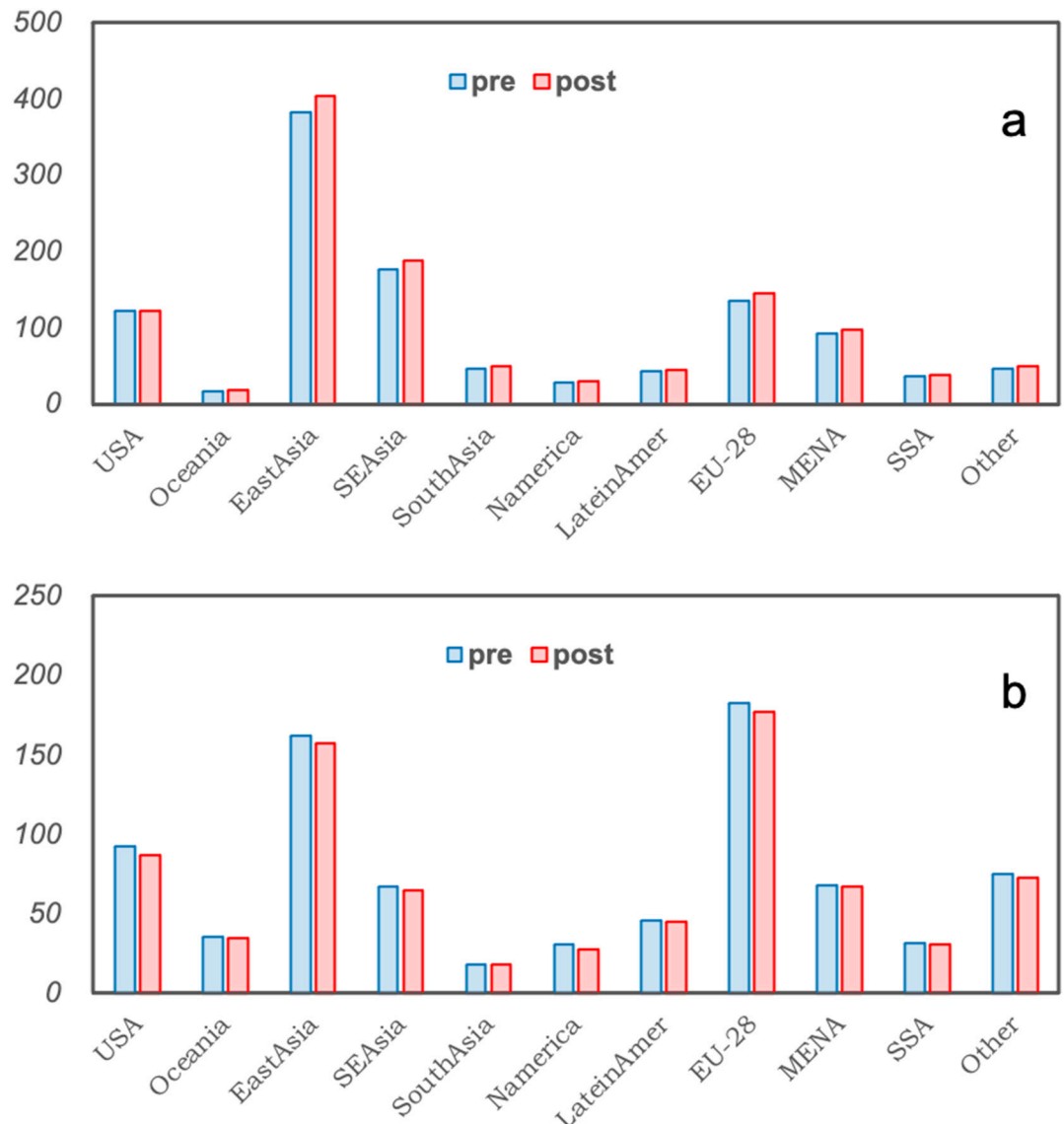

**Figure 4.** Spatial distribution of emissions included in China's trade (MtCO$_2$), where (**a**) represents emissions of exports and (**b**) represents emissions of imports.

### 4.2. Emissions Characteristics of China's Net Exports

As the "factory of the world", China has always been an export-oriented country, meaning that its carbon emissions from exports are higher than those from imports from other countries. Based on this, we calculate China's net emissions from international trade based on the previous results. As shown in Figure 5, China exports a large amount of carbon emissions in industries with high energy consumption, such as nonferrous metals, minerals, coal, and petroleum gas production. Excluding a few major sectors, the net emissions of other sectors are much lower. This situation is related to the long-term economic growth mode of the Chinese government. The government has invested heavily in energy-intensive industries to drive the rapid growth of the country's GDP. However, this situation is currently improving. With the adjustment of national strategies, environmental governance has been given equal importance to economic growth. Green sustainable development and the ensuing energy consumption revolution both reflect the Chinese government's determination to adapt to climate change. China is determined to start from multiple angles to resolve the contradiction between trade development and emissions growth.

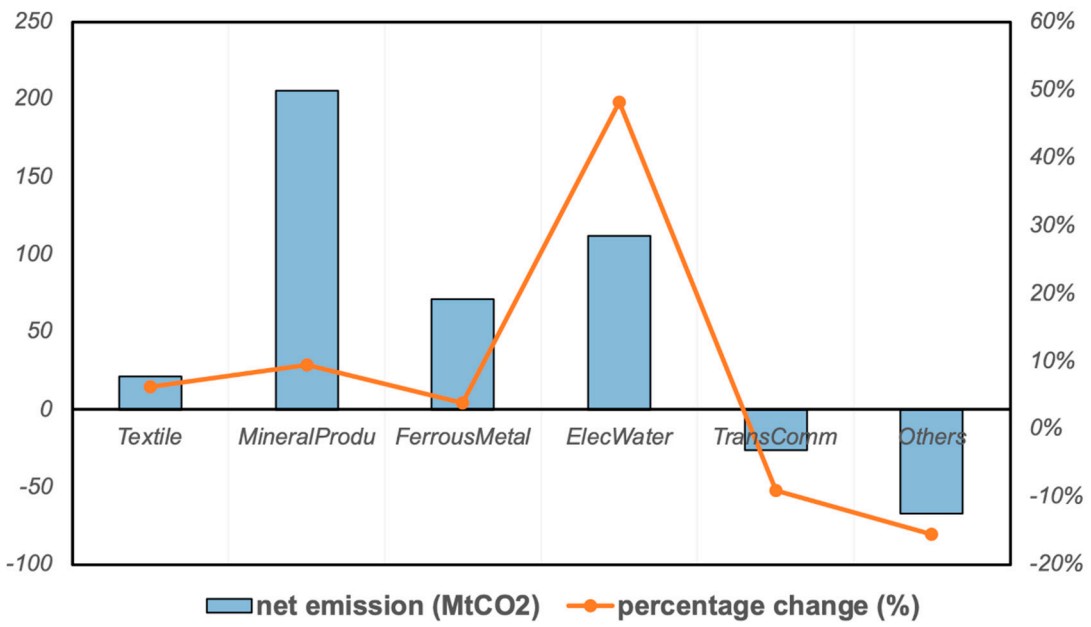

**Figure 5.** China's net export emissions and percentage change.

### 4.3. The Impact of the Trade Conflict on Climate Change

The most direct impact of human activities on climate change occurs through increases or decreases in carbon dioxide emissions. Based on the above results, China's carbon emissions in international trade have not changed much in the short term after the start of the trade conflict. Although import emissions have decreased, emissions from exports, the main component of China's trade, have not decreased but rather have increased. However, from a long-term perspective, the impact of the trade conflict on China's adaptation to climate change is likely to be more pronounced.

First, the trade conflict between China and the United States has had an impact not only on trade but also on the social economy of the two countries. The Chinese economy is in a "new normal" phase, the transitional stage from extensive growth based on scale and speed to intensive growth based on quality and efficiency. The negative impact of the trade conflict on China's economy is bound to delay its progress. As mentioned above, China's main exports in international trade come from the massive output of its energy-intensive industries. To ensure the steady development of the domestic economy and eliminate the negative effects of the trade conflict, government investment in these industries is not likely to change significantly.

Second, China has recently put forward a goal of achieving carbon neutrality by 2060. This plan is closely related to China's abundant wind power, hydropower, geothermal, and other new energy potential. China's abundant natural resources make it possible to achieve this goal. However, the new energy industry has a great demand for technology and equipment. China's current level of technology is not adequate to support independent achievement of its objectives. The import of technology and equipment is thus vital to the development of new energy. The trade conflict between China and the United States is set to have an impact on China's imports in and slow down the development of its domestic new energy industry, such that more effort will be required to achieve green development goals such as carbon neutrality.

Finally, as the economy develops, China's energy dependence will increase. Although China has a large amount of low-cost coal resources, considering the concept of green development, coal energy must gradually be replaced. On the other hand, China will have a higher degree of dependence on oil and natural gas, which are not abundant in the country, and thus will face much external uncertainty. In the international energy market, the United States will become a major oil and gas exporter in the market with the realization

of its shale gas revolution and energy independence strategy. In the face of China's massive natural gas demand, energy cooperation between China and the United States may offer a new opportunity to improve the trade imbalance between the two countries.

## 5. Conclusions

The trade conflict between China and the United States has had an impact on China's import and export markets, which in turn has affected the carbon emissions contained in China's imports and exports in international trade and will affect China's response to climate change. In China–US trade, the trade volume of goods subject to tariffs has been greatly reduced, while in other sectors, import emissions have increased and export emissions have decreased. For the global market, China's export emissions to the rest of the world have increased slightly, while import emissions have decreased slightly. The trade conflict will cause China's net export emissions to continue to increase, with the change concentrated in energy-intensive industries.

At the same time, it can be seen that although the trade share between China and the United States is not large in comparison with the world total, some of the sectors involved in the trade war are the main sectors involved in trade between the two countries, and they all contribute a large share to China's total trade volume. The sharp decline in trade in these sectors will also have impacts and raise opportunities in China's inland markets. On the other hand, the trade conflict will affect China's social economy from other angles in the long run as well as some of China's strategies to adapt to climate change. Whether through a negative impact on the domestic economy or restricted imports of technology and equipment, the trade conflict will slow down the development of China's new energy industry. The energy trade may provide an opportunity to solve the problem of the trade imbalance between the two countries.

This paper still has many shortcomings, especially in terms of data. On the one hand, due to the difficulty of obtaining data from all countries, we assume that the regional emission intensity is consistent, and there will be considerable uncertainty; on the other hand, we have also simplified the additional levy departments when it comes to tariff plans, due to the GTAP model. It is not easy to completely match the actual situation. We selected key departments to impose tariffs and simulate.

**Author Contributions:** Conceptualization, J.C. and W.D.; methodology, F.Y.; software, F.Y.; validation, F.Y. and Z.W.; formal analysis, Z.W.; resources, J.C.; data curation, F.Y.; writing—original draft preparation, F.Y.; writing—review and editing, J.C.; visualization, F.Y.; supervision, J.C.; project administration, J.C. All authors have read and agreed to the published version of the manuscript.

**Funding:** This research was funded by the National Key Research and Development Program of China (2018YFC1509003, 2016YFA0602703) and the National Natural Science Foundation of China (42075167).

**Data Availability Statement:** Not applicable.

**Acknowledgments:** The authors sincerely thank State Key Laboratory of Earth Surface Processes and Resource Ecology for supporting this research.

**Conflicts of Interest:** The authors declare that they have no known competing financial interests or personal relationships that could have appeared to influence the work reported in this paper.

## Appendix A

**Table A1.** Category of countries.

| Region Abbreviations | Comprising | Description |
|---|---|---|
| China | chn | China |
| USA | usa | USA |
| Oceania | aus, nzl, xoc | Oceania |
| EastAsia | hkg, jpn, kor, mng, twn, xea, brn | East Asia (Except China) |
| SEAsia | knm, idn, lao, mys, phl, sgp, tha, vnm, xse | Southeast Asia |
| Namerica | can, mex, xna | North America |
| LatinAmer | arg, bol, bra, chl, col, ecu, pry, per, ury, ven, xsm, cri, gtm, hnd, nic, pan, slv, xca, dom, jam, pri, tto, xcb | Latin Amercia |
| EU_28 | aut, bel, bgr, hrv, cyp, cze, dnk, est, fin, fra, deu, grc, hun, irl, ita, lva, ltu, lux, mlt, nld, pol, prt, rou, svk, svn, esp, swe, gbr | European Union 28 |
| MENA | bhr, irm, isr, jor, kwt, omn, qat, sau, tur, are, xws, egy, mar, tun, xnf | Middle East and North Africa |
| SSA | ben, bfa, cmr, civ, gha, gin, nga, sen, tgo, xwf, xcf, xac, eth, ken, mdg, mwi, mus, moz, rwa, tza, uga, zmb, zwe, xec, bwa, nam, zaf, xsc | Sub-Saharan Africa |
| RestofWorld | che, nor, xef, alb, blr, rus, ukr, xee, xer, kaz, kgz, tjk, xsu, arm, zae, geo, xtw | Rest of World |

**Table A2.** Category of sectors.

| Sectors Reclassified | Sectors in GTAP | Sectors in China | Category |
|---|---|---|---|
| GrainsFesFis | Grain, Fes, Fis | Farming, Forestry, Animal Husbandry, Fishery, and Water Conservancy | Primary industry |
| Coal | Coal | Coal Mining and Dressing | Energy production |
| OilGas | Oil, Gas | Petroleum and Natural Gas Extraction | Energy production |
| OtherMineral | Mineral | Ferrous Metals Mining and Dressing, Nonferrous Metals Mining and Dressing, Nonmetal Minerals Mining and Dressing, Other Minerals Mining and Dressing | Energy production |
| ProcFood | Food Production | Food Processing, Food Production | Light industry |
| BeveragesTob | Beverage production, Tobacco Production | Beverage Production, Tobacco Processing | Light industry |
| Textile | Textile | Textile Industry | Light industry |
| Wearing | Wearing | Garments and Other Fiber Products | Light industry |
| LeatherProd | Leather Production | Leather, Furs, Down, and Related Products | Light industry |
| WoodProduct | Wood Production | Logging and Transport of Wood and Bamboo, Timber Processing, Bamboo, Cane, Palm Fiber and Straw Products | Light industry |
| PaperProduct | Paper Production | Papermaking and Paper Products | Light industry |
| Transport | Transport Equipment | Transportation Equipment | Light industry |
| MetalProduct | Metal Production | Metal Products | Heavy industry |
| OthLightMnfc | Light Manufacture | Furniture Manufacturing, Printing and Record Medium Reproduction, Cultural, Educational and Sports Articles | Light industry |
| PetroleumCoa | Petroleum, Coal production | Petroleum Processing and Coking, Raw Chemical Materials, and Chemical Products | Energy production |
| ChemicalPro | Chemical Production | Chemical Fiber | Heavy industry |
| BasicPharmac | Basic Pharmacy | Medical and Pharmaceutical Products | Light industry |
| RubberPlasti | RubberPlastic | Rubber Products, Plastic Products | Heavy industry |
| MineralProdu | Mineral Production | Nonmetal Mineral Products | Heavy industry |
| FerrousMetal | Ferrous Metal Production | Smelting and Pressing of Ferrous Metals | Heavy industry |
| OtherMetal | Other Metal Production | Smelting and Pressing of Nonferrous Metals | Heavy industry |
| ElectronicEq | Electronic Equipment | Electric Equipment and Machinery | Electric Equipment and Machinery |

**Table A2.** *Cont.*

| Sectors Reclassified | Sectors in GTAP | Sectors in China | Category |
|---|---|---|---|
| ElectricalEq | Electrical Equipment | Electronic and Telecommunications Equipment, Instruments, Meters, Cultural and Office Machinery | Electric Equipment and Machinery |
| OthHeavyMnfc | Other Heavy Manufacture | Ordinary Machinery, Equipment for Special Purposes, Instruments, Meters, Cultural and Office Machinery, Other Manufacturing Industry | Heavy industry |
| ElecWater | Electricity, Water | Production and Supply of Electric Power, Steam and Hot Water, Production and Supply of Tap Water | Energy production |
| GasManufactu Constructio | Gas Manufacture Construction | Production and Supply of Gas Construction | Energy production Construction |
| TransComm | Trans Commerce | Transportation, Storage, Post and Telecommunication Services | Services industry |
| OthServices | Other Services | Wholesale, Retail Trade and Catering Services, Others | Services industry |

**Table A3.** Emission factors of each type of fuels.

| No. | Fuels in China's Energy Statistics | Fuels in This Study | $NCV_i \times CC_i$ (t C/$10^4$ ton) |
|---|---|---|---|
| 1 | Raw coal | Raw coal | 5.5272 |
| 2 | Cleaned coal | Cleaned coal | 6.8432 |
| 3 | Other washed coal | Other washed coal | 3.948 |
| 4 | Briquettes | Briquette | 4.7376 |
| 5 | Gangue Coke | Coke | 8.7864 |
| 6 | Coke oven gas | Coke over gas | 34.5989 |
| 7 | Blast furnace gas Converter gas Other gas | Other gas | 17.8367 |
| 8 | Other coking products | Other coking products | 7.686 |
| 9 | Crude Oil | Crude oil | 8.6344 |
| 10 | Gasoline | Gasoline | 8.316 |
| 11 | Kerosene | Kerosene | 8.624 |
| 12 | Diesel oil | Diesel oil | 8.686 |
| 13 | Fuel oil | Fuel oil | 9.073 |
| 14 | Naphtha Lubricants Paraffin White spirit Bitumen asphalt Petroleum coke Other petroleum products | Other petroleum products | 8.772 |
| 15 | Liquefied petroleum gas (LPG) | LPG | 9.4 |
| 16 | Refinery gas | Refinery gas | 8.686 |
| 17 | Nature gas | Nature gas | 59.5948 |

There are 26 kinds of fossil fuels in China's energy statistics system. Because the quality of some fuels is similar to that of other fuels, this paper combines these fuels into 17 types. Among the 17 types of fossil fuels, raw coal, crude oil, and natural gas are the main energy sources, and the other 14 fuels are classified as secondary energy sources.

## Appendix B

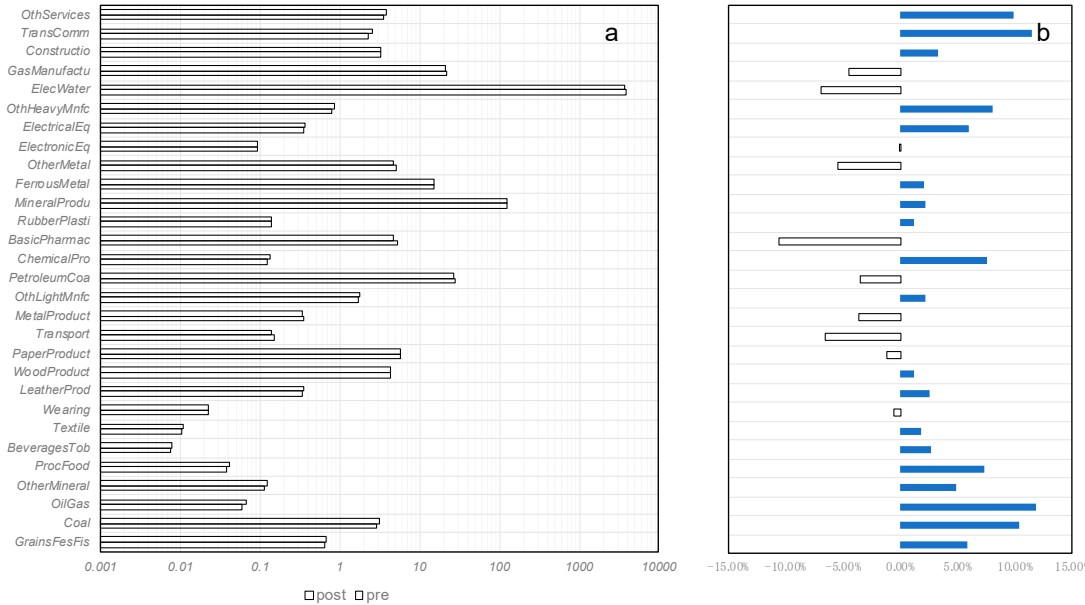

**Figure A1.** Changes in the world's carbon emissions from China's exports, where (**a**) represents the change in carbon emissions (MtCO$_2$) and (**b**) represents the percentage change.

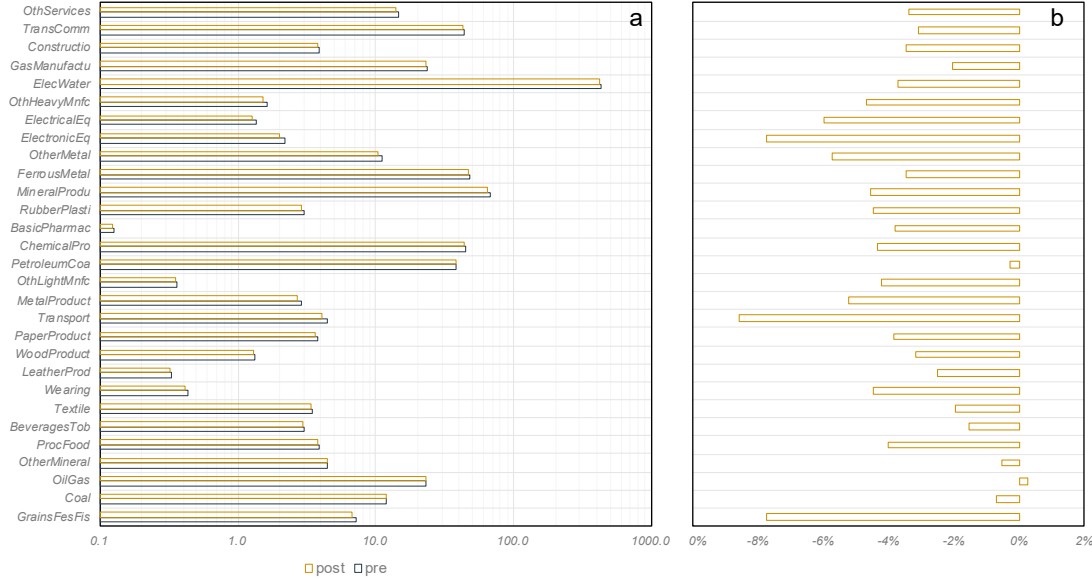

**Figure A2.** Changes in the world's carbon emissions from China's imports, where (**a**) represents the change in carbon emissions (MtCO$_2$) and (**b**) represents the percentage change.

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
