# Peer review of "The Impact on Carbon Emissions of China with the Trade Situation versus the U.S."

_sustainability, doi:10.3390/su131810324_

Round 1
Reviewer 1 Report
1. This manuscript appears to be looking at changes in emissions due to the China-US trade conflict that are generated by Chinese producers as a result of exports by China, and changes in emissions generated by producers in other countries as a result of China's imports of goods from those countries. The manuscript does not appear to be looking at changes in emissions due to changes in Chinese production of goods that are consumed within China, or changes in emissions due to changes in production of goods in other countries that are consumed within those countries, or changes in emissions due to changes in exports and imports not involving China (e.g. the US and Japan). This means that the manuscript does not present a complete assessment of the impact of the China-US trade conflict on global greenhouse emissions or even China's greenhouse gas emissions. Instead, it just focuses on emissions embodied in China's exports and imports. This raises the question of why this partial analysis of changes in greenhouse gas emissions is interesting or relevant, especially when the GTAP model could be used in support of a complete assessment. The authors need to explain why their results are interesting and important.
2. The manuscript is missing details about the tariffs imposed by the Trump administration on China, such as timing (there were multiple rounds of tariffs), tariff rates over time, and industries targeted. The manuscript would benefit from a table showing tariff rates by industry/product at the point in time selected by the authors for their analysis.
3. The manuscript would benefit from a more detailed description of the GTAP model, especially how international trade is modeled.
4. The assumption that countries (other than China?) have the same emission coefficients (line 100) is a strong assumption. The authors should briefly discuss the sensitivity of their results to this assumption.
5. It was unclear to me why China's net emissions to the world as a whole have increased as a result of the China-US trade conflict. This needs a better explanation.
6. For CO2 emissions (lines 148-149) do you mean just CO2 or CO2 equivalents, including other greenhouse gases?
Minor Comments:
7. The first paragraph in section 1 makes the manuscript feel dated ... the information there will soon be out of date. It would be best to either remove the paragraph or write it in a way that will hold up over time.
8. Of all the equations in the GTAP model, why have the authors chosen to present the government utility function in equation (1)?
9. P^EX in equation (2) is not defined.
10. What is the "sim" line in Figures 1a and 1b?
11. In figures 2b and 3b, I think you mean the percentage change rather than the rate of change.
Author Response
Response to reviewer 1:
Point 1: This manuscript appears to be looking at changes in emissions due to the China-US trade conflict that are generated by Chinese producers as a result of exports by China, and changes in emissions generated by producers in other countries as a result of China's imports of goods from those countries. The manuscript does not appear to be looking at changes in emissions due to changes in Chinese production of goods that are consumed within China, or changes in emissions due to changes in production of goods in other countries that are consumed within those countries, or changes in emissions due to changes in exports and imports not involving China (e.g. the US and Japan). This means that the manuscript does not present a complete assessment of the impact of the China-US trade conflict on global greenhouse emissions or even China's greenhouse gas emissions. Instead, it just focuses on emissions embodied in China's exports and imports. This raises the question of why this partial analysis of changes in greenhouse gas emissions is interesting or relevant, especially when the GTAP model could be used in support of a complete assessment. The authors need to explain why their results are interesting and important.
Response 1: We thank the Reviewer for careful reading of our paper and for the helpful comments. The impact of trade policy on the domestic economy is far less than that of some domestic policies or global emergencies, so we believe that more impact is concentrated on import and export commodities, so we focused on the analysis of these changes. Trade accounts for an increasing trend in the total emissions of economic sectors. This part of the research on emissions is meaningful, and tariff changes caused by changes in trade directly affect the volume of goods and the emissions contained in it. We want to observe how the emissions will change from this perspective. This should be meaningful.
Point 2: The manuscript is missing details about the tariffs imposed by the Trump administration on China, such as timing (there were multiple rounds of tariffs), tariff rates over time, and industries targeted. The manuscript would benefit from a table showing tariff rates by industry/product at the point in time selected by the authors for their analysis.
Response 2: We thank the Reviewer for the comment, details about the tariffs imposed by the Trump administration on China can be found from “Reference 7”. After that, We selected some departments that are mainly involved in the levy of tariffs and matched the GTAP model for simulation.
Point 3: The manuscript would benefit from a more detailed description of the GTAP model, especially how international trade is modeled.
Response 3: Thank the Reviewer for valuable advice, we have added some descriptions of the trade part in model.
See new L133-142.
Point 4: The assumption that countries (other than China?) have the same emission coefficients (line 100) is a strong assumption. The authors should briefly discuss the sensitivity of their results to this assumption.
Response 4: Thank the Reviewer for valuable advice, we assume that the emission coefficients of commodities in each regions is the same, we have modified the sentence. Here we adopt this method because the statistical data of all countries is not easy to obtain, so we have made such an assumption. Therefore, we added some discussion about the uncertainty of the data results at the end of the paper.
Point 5: It was unclear to me why China's net emissions to the world as a whole have increased as a result of the China-US trade conflict. This needs a better explanation.
Response 5: We thank the Reviewer for the comment, we think this issue needs to be explained separately from the import and export aspects. The first is the decline in imports, which is the result of rising consumer product prices. As for exports, China is still more suitable in the international market, and exports will be compensated to other countries. With less imports and less changes in exports, net emissions increase.
Point 6: For CO2 emissions (lines 148-149) do you mean just CO2 or CO2 equivalents, including other greenhouse gases?
Response 6: We thank the Reviewer for the comment, CO2 emissions in this paper mean just CO2.
Minor Comments:
Point 7: The first paragraph in section 1 makes the manuscript feel dated ... the information there will soon be out of date. It would be best to either remove the paragraph or write it in a way that will hold up over time.
Response 7: We thank the Reviewer for the comment, we have modified some sentences in the first paragraph. We believe that China is one of the most important importers of goods in the United States, and once there are problems in other importing countries, China-US trade relations will change. The method of studying the changes in existing emissions can also be applied to changes in future trade conditions, which is also one of the meanings of this paper. So we want to keep this part of the text to illustrate that this paper may also be useful for reference in the future.
Point 8: Of all the equations in the GTAP model, why have the authors chosen to present the government utility function in equation (1)?
Response 8: We thank the Reviewer for the comment, we have modified the sentence.
See new L132.
Point 9: P^EX in equation (2) is not defined.
Response 9: We thank the Reviewer for the comment, we have added related definition.
See new L150.
Point 10: What is the "sim" line in Figures 1a and 1b?
Response 10: We thank the Reviewer for the comment, “sim” means the percentage change of carbon emissions, we have modified the figure.
See new L268.
Point 11: In figures 2b and 3b, I think you mean the percentage change rather than the rate of change.
Response 11: Following the Reviewer's remark, we have modified the figure.
See new L341.
Reviewer 2 Report
This is a very interesting study that, if viewed in general terms provides a useful insight into the impacts of the trade embargo's with China, and by China. The limiting factor in the study is "This paper uses a recursive method to project the 2014 data in the model to 2018". It is to be expected the more recent data would vary greatly the 2014 data, partcilcularly with the demand for iron ore growing as the basic ingredient for steel production. The consequence is a different emissions profile than is presented here. Even so, the research methodology is sufficiently sound to produce modeling that can be further developed ans a sound basis for other research.
Author Response
We thank the Reviewer for careful reading of our paper and for the helpful comments, we will try to find better solutions of the data in the future.
Reviewer 3 Report
- Why did you consider carbon emissions and not CO2 eq. ?
- I do not totally agree with this sentence "The LCA method is typically used for relatively simple and traceable inspections of production chains such as households and enterprises"
- LCA is a standardized methodology, what about IOA?
- Did you perform a sensitivity analysis?
- Fig1 "before (a) and after (b) ..."
- Everywhere (text and figures) CO2 should be CO2
Author Response
Response to Reviewer 2:
Comments and Suggestions for Authors
Point 1: Why did you consider carbon emissions and not CO2 eq. ?
Response 1: We thank the Reviewer for the comment , we did not find a good way to measure the emissions of other greenhouse gases contained in traded goods, so we selected CO2 emissions as the research content.
Point 2: I do not totally agree with this sentence "The LCA method is typically used for relatively simple and traceable inspections of production chains such as households and enterprises"
Response 2: We thank the Reviewer for the comment , we think that Life cycle assessment(LCA) is a bottom-up methodology. It is not easy to use this method to calculate emissions transfers between many industries.
Point 3: LCA is a standardized methodology, what about IOA?
Response 3: We thank the Reviewer for the comment , we think that IOA can solve some of LCA's shortcomings. Despite the fact that LCA is a powerful tool for evaluating the embodied energy, some inputs are inevitably neglected and truncated after several stages, due to the time-consuming and infinite trace process. Instead, IOA can provide a panorama of energy flows of the entire system and consistent energy intensity data, and can be docked with GTAP model.
Point 4: Did you perform a sensitivity analysis?
Response 4: We thank the Reviewer for the comment , we added some discussion about the uncertainty of the data results at the end of the paper.
Point 5: Fig1 "before (a) and after (b) ..."
Response 5: We thank the Reviewer for the comment , in Fig.1, (a) represents emissions from China to the United States and (b) represents emissions from the Unit-ed States to China. Do you mean we should use “(a)” replace “a”?
Point 6: Everywhere (text and figures) CO2 should be CO2
Response 6: Thank the Reviewer for valuable advice, we have replaced the word.
See new L172, 269, 290, 293, 322, 429, 432.
Round 2
Reviewer 1 Report
This manuscript is improved from its previous version. I have just a few minor comments:
1. The manuscript should state explicitly that CO2 is referring only to carbon dioxide, because many other studies use CO2 to refer to both carbon dioxide and CO2 equivalents of other greenhouse gases.
2. I still feel that the first paragraph of the manuscript (about Covid) is a distraction from the theme of the paper and will make the manuscript feel out-of-date very soon.
3. There are a few places where the authors use "rate of change" where I think they mean "percentage change".
Author Response
Response to Reviewer 1:
This manuscript is improved from its previous version. I have just a few minor comments:
Point 1: The manuscript should state explicitly that CO2 is referring only to carbon dioxide, because many other studies use CO2 to refer to both carbon dioxide and CO2 equivalents of other greenhouse gases.
Response 1: We thank the Reviewer for the comment, we have added a description of CO2 data.
See L96-97.
Point 2: I still feel that the first paragraph of the manuscript (about Covid) is a distraction from the theme of the paper and will make the manuscript feel out-of-date very soon.
Response 2: We thank the Reviewer for the comment, we think that this part can help readers understand the uncertainty of trade relations. Perhaps this part of the content will out-of-date, but the scientific issues that require quantitative analysis that extend from these events will not out-of-date.
Point 3: There are a few places where the authors use "rate of change" where I think they mean "percentage change".
Response 3: We thank the Reviewer for the comment, we have replaced the words.
See L243, 251, 258, 261, 307, 388, 391.